

# Interplay between epigallocatechin-3-gallate and ionic strength during amyloid aggregation

Mantas Ziaunys, Kamile Mikalauskaite, Andrius Sakalauskas and Vytautas Smirnovas

Institute of Biotechnology, Life Sciences Center, Vilnius University, Vilnius, Lithuania

## ABSTRACT

The formation and accumulation of protein amyloid aggregates is linked with multiple amyloidoses, including neurodegenerative Alzheimer's or Parkinson's disease. The mechanism of such fibril formation is impacted by various environmental conditions, which greatly complicates the search for potential anti-amyloid compounds. One of these factors is solution ionic strength, which varies between different aggregation protocols during *in vitro* drug screenings. In this work, we examine the interplay between ionic strength and a well-known protein aggregation inhibitor—epigallocatechin-3-gallate. We show that changes in solution ionic strength have a major impact on the compound's inhibitory effect, reflected in both aggregation times and final fibril structure. We also observe that this effect is unique to different amyloid-forming proteins, such as insulin, alpha-synuclein and amyloid-beta.

## INTRODUCTION

Protein aggregation into amyloid fibrils is associated with the onset and progression of many amyloidoses (*Baker & Rice, 2012*), including widespread neurodegenerative disorders, such as Alzheimer's or Parkinson's disease (*Knowles, Vendruscolo & Dobson, 2014*; *Chiti & Dobson, 2017*). Countless studies and years of research have resulted in very few disease-modifying drugs (*Maurer et al., 2018*; *Park et al., 2020*), with most potential aggregation-inhibiting compounds failing at various stages of clinical trials (*Mehta et al., 2017*; *Cummings et al., 2020*), which, in turn, has not allowed to slow down the ever-increasing occurrence of amyloid-related disorders and even more cases are projected for the upcoming decades (*Hebert et al., 2013*; *Arthur et al., 2016*). This makes it vitally important to obtain a better understanding of protein aggregation (*Cohen et al., 2012*), as well as their interaction (*Fusco et al., 2018*) with drug molecules. Currently, there are more than a hundred compounds which have shown potential in having either a direct or indirect effect on the appearance and accumulation of these disease-related protein/peptide aggregates (*Williams, Sorribas & Howes, 2011*). It was also shown that there are multiple modes of interaction between potential drug molecules and fibrils,

Corresponding author
Mantas Ziaunys,
mantas.ziaunys@gmc.vu.lt

which include binding along their surface, inside cavities or by interacting with specific amino acids (*Landau et al., 2011*).

One of the main problems during anti-amyloid compound screenings is the impact of environmental conditions on both the aggregation rate and mechanism, as well as protein interaction. Multiple factors, such as protein concentration (*Sakalauskas, Ziaunys & Smirnovas, 2019*), temperature (*Tanaka et al., 2006*; *Colby et al., 2009*), agitation (*Petkova et al., 2005*), ionic strength (*Bousset et al., 2013*), pH (*Sneideris et al., 2015*) or denaturant concentration (*Colby et al., 2009*) can influence the initial amyloidogenic protein state (*Nettleton et al., 2000*), the course and rate of aggregation (*Morel et al., 2010*; *Noormägi et al., 2015*) and the resulting fibril conformation/morphology (*Morel et al., 2010*; *Zidar & Merzel, 2011*; *Nicoud et al., 2015*). Taking into consideration that amyloid-inhibitor interactions may be highly-selective (*Wang, Dong & Sun, 2012*; *Zhuang et al., 2016*; *Liu et al., 2018*; *Fusco et al., 2018*), distinct fibrillization intermediates or fibrils can have specific affinity to the drug molecules in question. This effect was demonstrated during alpha-synuclein and epigallocatechin-3-gallate (EGCG) experiments, where it was observed that changes to the solution's pH value or sample agitation drastically improve or impair the molecule's effectiveness at inhibiting alpha-synuclein aggregation (*Sneideris et al., 2019*; *Sternke-Hoffmann et al., 2020*).

One of these factors, which requires a more in-depth analysis, is ionic strength. It has been demonstrated that the solution's ionic strength (usually modulated by the addition of NaCl (*Jain & Udgaonkar, 2010*)) can influence the rate of aggregation by altering electrostatic interactions between protein molecules (*Ziaunys, Sneideris & Smirnovas, 2018*). It was also shown that this factor can determine the conformation and morphology of fibrils that form during aggregation (*Bousset et al., 2013*; *Gaspar et al., 2020*). In addition, ionic strength can influence compound interactions with fibrils and even cause conformation changes to preformed aggregates (*Mikalauskaite et al., 2020*). Taking all of this into account, it appears that ionic strength plays an important role in multiple stages of the amyloid formation process and could significantly alter experimental results and their conclusions regarding anti-amyloid compounds. Since there are many groups screening for potential aggregation-inhibiting drugs (*Findeis, 2000*; *Huang, Chao & Hu, 2020*) and each one has their own preferred/optimized fibrillization protocols (*Giorgetti et al., 2018*), there is a substantial number of different ionic strength conditions used. Considering the aforementioned effects that this factor may have on amyloid formation, as well as the possible highly-selective nature of anti-amyloid compounds, it is possible that certain potent drugs may appear ineffective under specific ionic strength conditions, leading to false-negative results.

In order to examine the magnitude of the effect that ionic strength has on anti-amyloid drug screening, one of the most well-known and commonly used compounds was chosen—epigallocatechin-3-gallate (EGCG). It has been shown on numerous studies, both *in vitro* and *in silico* (*Ngo et al., 2017*; *Tavanti, Pedone & Menziani, 2020*), that EGCG can alter the aggregation pathway of amyloidogenic proteins, leading to either off-pathway structures or slowing down the rate of fibril formation (*Andrich & Bieschke, 2015*). It is also known that this polyphenolic molecule undergoes autoxidation at neutral pH, which

increases its anti-amyloid potency and minimizes further structural changes during the aggregation reaction (*An, Feng & Zeng, 2017*; *Sneideris et al., 2019*).

In this work, the inhibitory effect of preoxidized EGCG was determined under a range of ionic strength conditions (below and above the typical physiological ionic strength; *Kawai, Wray & Güth, 1990*), using two neurodegenerative-disease-related proteins/peptides—amyloid-beta (Alzheimer's disease) and alpha-synuclein (Parkinson's disease), as well as insulin, which is associated with localized injection site amyloidosis (*Knowles, Vendruscolo & Dobson, 2014*), but is widely used as a model protein in amyloid studies. Due to the distinct amino acid sequences and aggregation conditions of the selected proteins/peptides, it was possible to gain a deeper insight into the effect that this factor has on EGCG inhibitory potential. We show how the interplay between ionic strength and EGCG has a unique effect on each protein/peptide aggregation kinetics and resulting fibril structures.

## MATERIALS AND METHODS

### Epigallocatechin gallate preparation

Epigallocatechin-3-gallate (Cat. No. M01719; Fluorochem, Derbyshire, UK) was dissolved in a 100 mM potassium phosphate (pH 7.4) buffer to a final concentration of 10 mM. Due to EGCG undergoing autoxidation at neutral pH under two of the three tested aggregation conditions, which changes the efficiency of its inhibition (*Sneideris et al., 2019*; *Wei et al., 2016*), the compound was preoxidized before being used in aggregation experiments by distributing the solution into 1.5 mL test-tubes (1 mL final volume) and incubating at 60 °C for 72 h without agitation (*Ziaunys et al., 2021a*). The solution's absorbance spectra were scanned every 24 h using a Shimadzu UV-1800 spectrophotometer in the range from 240 nm to 600 nm (1 nm steps, 3 mm pathlength cuvette) and baseline corrected. Prior to measurements, an aliquot of the solution was diluted 100 times by its initial reaction buffer. No significant changes to the absorbance spectra were observed after the 72 h mark, indicating that this time frame is sufficient for a complete oxidation process (Fig. S1). The final solution was then filtered through a 0.22 µm syringe filter and stored at 4 °C. Only the oxidized form of EGCG was used in all further experiments.

### Aggregation reactions

Human recombinant insulin powder (Cat. No. 91077C; Sigma–Aldrich, St. Louis, MO, USA) was dissolved in 20% acetic acid (pH ~ 1.8, (*Foderà, Van De Weert & Vestergaard, 2010*)) solutions, containing a range of NaCl concentrations (from 100 mM to 800 mM) to a final protein concentration of 400 µM ($\varepsilon_{280}$ = 6,335 $M^{-1}$ $cm^{-1}$). The insulin stock solutions were then combined with 10 mM ThT (Cat. No. T3516; Sigma-Aldrich, St. Louis, MO, USA) and 10 mM EGCG stock solutions and diluted using their respective 20% acetic acid solutions to a final protein concentration of 200 µM, 100 µM ThT and 25 µM EGCG. Control samples contained an equal volume of 100 mM potassium phosphate (pH 7.4) buffer solution in place of EGCG. The reaction solutions were then distributed into 96-well non-binding plates (Cat. No. 3881; Fisher Scientific, Pittsburgh, PA, USA, final volume 100 µL, six repeats for each condition), which were sealed using Nunc sealing-tape.

Aggregation kinetics were monitored at 60 °C without agitation, using a ClarioStar Plus plate reader (440 nm excitation and 480 nm emission wavelengths) with measurements taken every 5 min. Samples were placed in the 96-well plate in an alternating style (Fig. S2) to avoid any possible non-homogenous temperature variations throughout the plate having an effect on the average aggregation kinetics.

Alpha-synuclein was purified as described previously (*Šneideris et al., 2015*), lyophilized and stored at −20 °C. Prior to aggregation experiments, alpha-synuclein powder was dissolved in 20 mM potassium phosphate (pH 7.4) buffers containing a range of NaCl concentrations (from 100 mM to 800 mM) and filtered through a 0.22 μm syringe filter, after which the protein solution was diluted to 250 μM ($\varepsilon_{280}$ = 5960 $M^{-1}$ $cm^{-1}$). The alpha-synuclein stock solutions were then combined with 10 mM ThT and 10 mM EGCG stock solutions and diluted using their respective buffer solutions to a final protein concentration of 100 μM, 100 μM ThT and 100 μM EGCG. The reaction solutions were distributed to 96-well plates as described previously (final volume 80 μL, each well contained a 3 mm glass bead). Aggregation kinetics were monitored at 37 °C with constant 600 RPM orbital agitation.

Amyloid beta (1–42) was purified as described in Supplemental Material, stored on ice (~5 min) until its concentration was determined by integrating the chromatographic UV absorbance peak ($\varepsilon_{280}$ = 1,490 $M^{-1}$ $cm^{-1}$), after which it was immediately diluted with a 20 mM sodium phosphate (pH 6.3) buffer solution in a 1:2 ratio to yield a pH 7.0 peptide solution. Afterwards, it was further diluted using a 20 mM sodium phopshate (pH 7.0) buffer to reach a peptide concentration of 5 μM. The peptide solution was then combined with 20 mM sodium phosphate (pH 7.0) buffers, containing either 0 or 1.6 M NaCl, 10 mM ThT and 10 mM EGCG stock solutions to a final peptide concentration of 2 μM, 20 μM ThT, 25 μM EGCG and a range of NaCl concentrations (from 0 mM to 700 mM). All buffer, ThT and EGCG solutions were kept at 4 °C prior to use. The reaction solutions were then distributed to a 96-well plate as described previously (final volume 80 μL). Aggregation kinetics were monitored at 25 °C without agitation.

For all three proteins/peptides, the EGCG concentrations were initially optimized to result in a 3 to 4-fold increase in the relative $t_{50}$ values. The half-time ($t_{50}$), lag time ($t_{lag}$) and aggregation rate values were determined by fitting each curve with a Boltzmann sigmoidal equation as described previously (*Ziaunys et al., 2021b*).

## Fourier-transform infrared (FTIR) spectroscopy

Each condition samples were collected from their respective 96-well plates and combined to a final volume of ~0.6 mL (insulin fibrils) or ~0.48 mL (alpha-synuclein). The combined samples were centrifuged at 12,500 RPM for 20 min, after which the supernatant was removed. The fibril pellets were resuspended into 0.5 mL $D_2O$ with 400 mM NaCl (exchanging the solution to $D_2O$ removes the $H_2O$-specific absorbance in the Amide I region, while the addition of NaCl improves aggregate sedimentation (*Mikalauskaite et al., 2020*)). This centrifugation and resuspension procedure was repeated 4 times. After the final centrifugation, the fibril pellets were resuspended into 0.1 mL $D_2O$ without NaCl. Each sample was sonicated for 5 s using a Bandelin Sonopuls Ultrasonic homogenizer,

equipped with a MS-72 sonication tip (20% of maximum power, constant sonication). FTIR spectra were acquired as described previously (*Sneideris et al., 2019*) using a Bruker Invenio S FTIR spectrometer. A $D_2O$ spectrum was subtracted from each sample's spectrum, which were then normalized between 1,595 and 1,700 $cm^{-1}$. All data processing was done using GRAMS software. In the case of amyloid beta, the resulting spectra did not meet the quality necessary for an accurate analysis due to the significantly lower concentration of fibrils and the resulting low signal-to-noise ratio.

### ThT and EGCG absorbance assay

Insulin fibrils were prepared by incubating the previously described insulin aggregation reaction solutions in 2.0 mL non-binding test-tubes (2 mL final volume) at 60 °C without agitation for 24 h. The fibril samples were then centrifuged at 12,500 RPM for 20 min and resuspended into 2 mL of 20% acetic acid solutions, containing either 100 mM or 800 mM NaCl. This centrifugation and resuspension procedure was repeated 4 times. After this, both fibril samples were sonicated for 10 min on ice, using a MS-73 sonication tip (40% of maximum power, 30 s sonication/30 s rest intervals). The resulting sonicated fibril solutions were combined with 10 mM ThT and 10 mM EGCG stock solutions, resulting in a 2-fold diluted fibril sample, containing 0 µM, 50 µM or 100 µM ThT and 0 µM, 50 µM or 100 µM EGCG. For control samples, the fibril solution was replaced with a 20% acetic acid solution, containing either 100 mM or 800 mM NaCl.

All samples were then vigorously agitated and mixed for 10 s, incubated at room temperature without agitation for 20 min and centrifuged at 12,500 RPM for 20 min. From each sample, a portion of the supernatant was carefully removed. The supernatant was then scanned using a Shimadzu UV-1800 spectrophotometer in the wavelength range from 240 nm to 600 nm in a 10 mm pathlength cuvette. For each condition, three spectra were scanned, averaged and baseline corrected based on the absorbance value at 600 nm.

## RESULTS

The inhibitory potential of oxidized epigallocatechin-3-gallate (further referred to as EGCG) was first examined on insulin, under a range of ionic strength conditions. At 0 mM NaCl, insulin aggregation in the presence of EGCG did not complete in a reasonable timeframe (the aggregation extended into time frames where sample evaporation and dye stability became an issue). When the concentration of NaCl was low (100 mM), the aggregation half-time ($t_{50}$) had an average value of ~1,700 min in the presence of EGCG, while the control sample's $t_{50}$ was ~370 min (Fig. 1A). In this case, the relative $t_{50}$ value was ~4, indicating a strong aggregation-inhibiting effect (Fig. 1B). When the concentration of NaCl was increased to 200 mM, the relative $t_{50}$ value remained similar, however, both the control and EGCG samples aggregated significantly quicker. Further increase in ionic strength led to a considerable shift in the relative $t_{50}$ values (from ~4 to ~1.5–2), while the half-times of samples continued to decrease with rising NaCl concentrations. A one-way ANOVA Bonferroni means comparison analysis displayed a significant difference between the 100–200 mM and 300–800 mM group relative $t_{50}$ values ($p = 0.01$) with no significant variation within each group. The decreasing $t_{50}$ values also greatly

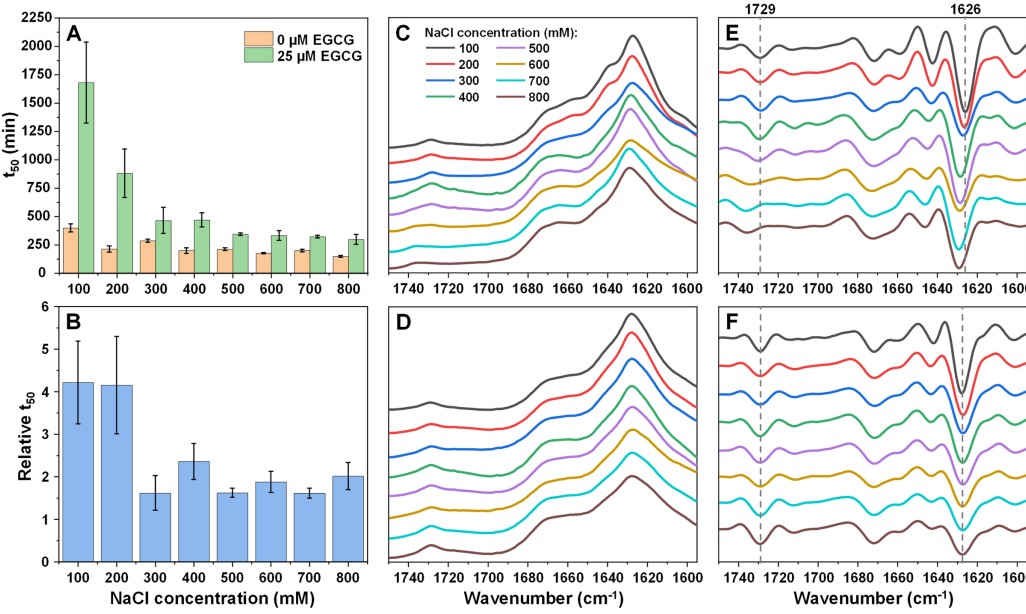

**Figure 1** Insulin (200 μM ) aggregation half-time ($t_{50}$) values and resulting fibril FTIR spectra. Insulin aggregation $t_{50}$ values in the absence or presence of 25 μM EGCG (A) and relative $t_{50}$ values (B). Relative $t_{50}$ values were obtained by calculating the ratio between average $t_{50}$ of samples with EGCG and average $t_{50}$ without EGCG. FTIR spectra of insulin fibrils prepared under a range of NaCl concentrations in the absence (C) or presence (D) of 25 μM EGCG and their second derivatives (E and F respectively). Dotted lines show second derivative positions, which experience variation between different spectra. For each condition, half-time values were calculated from six repeats, error bars are for one standard deviation. All raw kinetic and FTIR data is available as Supplemental Information.

reduced the stochasticity of spontaneous aggregation (Fig. S3), which was quite large at lower ionic strength conditions (likely due to a significantly longer lag time, coupled with the spontaneous nature of unseeded aggregation (*Foderà et al., 2008b*)). The lag time ($t_{lag}$) of all reactions followed a similar tendency as the $t_{50}$ values and the aggregation rate did not experience significant differences under all conditions (Fig. S6). Plotting a lag time dependence on the square root of NaCl concentration (*Zhou & Pang, 2018*; *Perez-Jimenez et al., 2004*) (Fig. S6) revealed that there was a discontinuity at 300 mM NaCl, after which the increasing ionic strength had a significantly lesser effect on the lag time.

The end-point fluorescence intensity values of fibril-bound ThT were not used as a means of identifying the inhibitory potential of EGCG for any of the three proteins/ peptides used in this study. This was due to the fact that exogenous compounds may cause ThT fluorescence intensity variations by direct interaction, inner filter or non-conjugated FRET effects (*Hudson et al., 2009*; *Ran et al., 2011*; *Ziaunys, Sakalauskas & Smirnovas, 2020*). Differences in fibril secondary structure or morphology may also alter its signal intensity (*Ziaunys, Sakalauskas & Smirnovas, 2020*; *Bousset et al., 2013*).

The resulting fibril secondary structures were analysed using FTIR spectroscopy in order to determine whether EGCG and changes in ionic strength had any influence on insulin aggregate conformations. When the reaction solutions did not contain EGCG, insulin fibrils appear to have undergone a structural transition between low and high NaCl

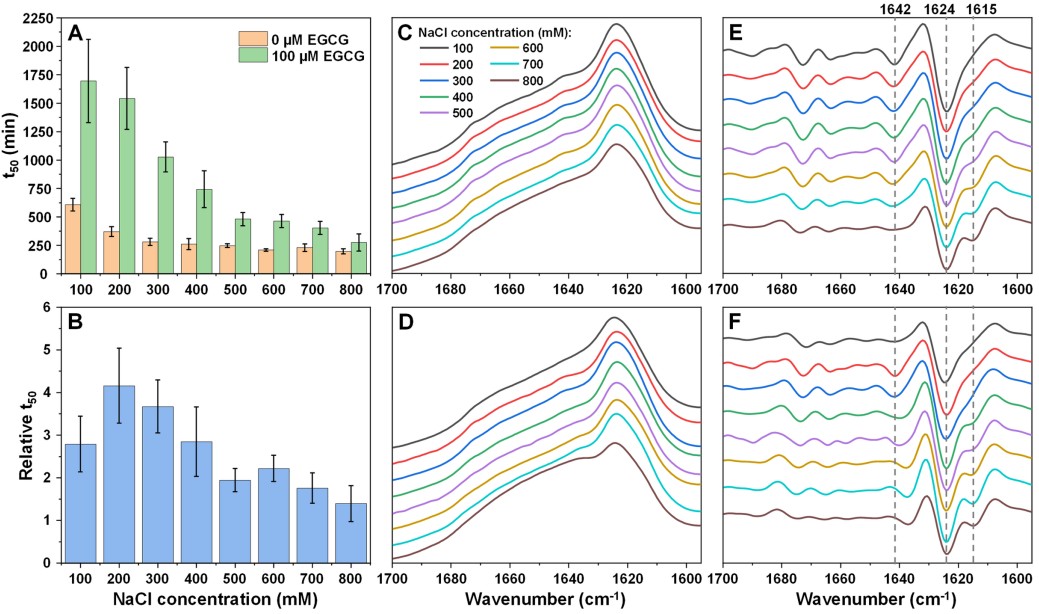

**Figure 2 Alpha-synuclein (100 μM) aggregation half-time ($t_{50}$) values and resulting fibril FTIR spectra.** Alpha-synuclein aggregation $t_{50}$ values in the absence or presence of 100 μM EGCG (A) and relative $t_{50}$ values (B). Relative $t_{50}$ values were obtained by calculating the ratio between average $t_{50}$ of samples with EGCG and average $t_{50}$ without EGCG. FTIR spectra of alpha-synuclein fibrils prepared under a range of NaCl concentrations in the absence (C) or presence (D) of 100 μM EGCG and their second derivatives (E and F respectively). Dotted lines show second derivative positions, which experience variation between different spectra. For each condition, half-time values were calculated from six repeats, error bars are for one standard deviation. All raw kinetic and FTIR data is available as Supplemental Information.

concentration conditions. At low ionic strength (100–200 mM NaCl), the FTIR spectra (Fig. 1C) main maxima are at 1,626 cm$^{-1}$, which is associated with the presence of beta-sheet hydrogen bonds (*Barth, 2007*). The second derivative spectra (Fig. 1E) also contain minima at 1,642 cm$^{-1}$ (weaker beta-sheet hydrogen bonds), 1,659 cm$^{-1}$ and 1,672 cm$^{-1}$ (turn/loop motifs), as well as 1,729 cm$^{-1}$ (deuterated carboxyl groups (*Surmacz-Chwedoruk et al., 2012*)). When ionic strength is higher, the main maximum shifts to 1,628–1,629 cm$^{-1}$, indicating weaker beta-sheet hydrogen bonding than in the lower ionic strength samples. The band associated with deuterated carboxyl groups (1,729 cm$^{-1}$) is also reduced at higher NaCl concentrations.

When the reaction solutions contained 25 μM EGCG, the main FTIR spectrum (Fig. 1D) maximum position is at 1,627–1,628 cm$^{-1}$ under all ionic strength conditions. Unlike insulin fibrils without EGCG, the spectra of fibrils, prepared at different NaCl concentrations, are very similar and only a slight decreasing of 1,642 cm$^{-1}$ band can be observed in second derivative spectra with increased ionic strength (Fig. 1F). This means that the presence of the inhibitor is also a factor in determining the conformation of the aggregate and it stabilises a certain secondary structure.

When alpha-synuclein was aggregated under a range of ionic strength conditions with and without EGCG, we observed a similar reduction in $t_{50}$ values with increasing NaCl concentration (Fig. 2A), as was the case for insulin aggregation. However, the change in

relative $t_{50}$ was quite different (Fig. 2B). When the solution's ionic strength was increased from 100 mM to 200 mM NaCl, the relative $t_{50}$ value increased, indicating a stronger inhibitory effect of EGCG. After this point, the value gradually decreased with rising NaCl concentrations, eventually becoming within margin of error to the control. A Bonferroni means comparison revealed that the relative $t_{50}$ values above 400 mM NaCl were significantly different ($p = 0.01$) from the 200 mM NaCl condition values. Unlike with insulin aggregation, there was no sudden shift in the relative $t_{50}$ values, however, the higher stochasticity at low ionic strength conditions was quite similar (Fig. S4). As with insulin at 0 mM NaCl conditions, alpha-synuclein did not aggregate in a reasonable timeframe when in the presence of EGCG (the aggregation began to occur only after ~2,000 min for samples without EGCG, which created the issue of sample evaporation and ThT instability (*Foderà et al., 2008a*)). The aggregation rate values were mostly within margin of error without any major deviations, while $t_{lag}$ values followed a similar tendency as $t_{50}$ values (Fig. S6). Unlike with insulin, a discontinuity in the lag time dependence on the square root of NaCl concentration only appeared at 500 mM NaCl (Fig. S6).

The FTIR spectra of alpha-synuclein fibrils prepared in the absence (Figs. 2C and 2E) or presence (Figs. 2D and 2F) of EGCG show a gradual conformational shift with increasing NaCl concentration, reflected by the appearance of a minimum in the second derivative spectra at 1,615 $cm^{-1}$, which is related to stronger hydrogen bonds in the beta-sheet structure. While the main maxima (and the main minima in the second derivative spectra) all share the same position (1,624 $cm^{-1}$, related to hydrogen bonds in the beta-sheet structure), the minimum at 1,642 $cm^{-1}$ (weak hydrogen bonds) disappears at the highest ionic strength conditions when there is no EGCG present in solution. When alpha-synuclein aggregates with the inhibitor, this minimum is almost undetectable at low ionic strength conditions (100 mM NaCl), then becomes visible (200–300 mM NaCl) and then shifts towards 1,637 $cm^{-1}$. EGCG also appears to cause minor variations in the region associated with turn/loop motifs (1,660–1,680 $cm^{-1}$) upon an increasing concentration of NaCl. Overall, the structural changes seem to be a lot less sudden than was the case with insulin fibrils, which falls in line with the gradual change to relative $t_{50}$ values.

In the case of amyloid beta (1–42), it was possible to track the impact of ionic strength on its aggregation and the effect of EGCG from 0 mM NaCl due to the relatively quick aggregation reaction times (Fig. S5). The first notable thing is that the $t_{50}$ values experienced a higher change when in the absence of EGCG, where they shifted from ~50 min at 0 mM NaCl to 20–30 min at higher NaCl concentrations (Fig. 3A). The $t_{50}$ values in the presence of EGCG only appeared to experience minimal variations. The relative $t_{50}$ values (Fig. 3B) did not follow any significant trend, as all values were mostly within margin of error. However, a Bonferroni means comparison analysis revealed that the 0 mM NaCl relative $t_{50}$ values were significantly different ($p = 0.01$) from the 300–600 mM NaCl condition values. Unlike with both other proteins, the inhibitory effect of EGCG did not undergo considerable changes, as even 700 mM NaCl conditions resulted in ~2.5 times higher $t_{50}$ values when compared to the control. The $t_{lag}$ and aggregation rate values also did not display any significant variations, however, the lag time
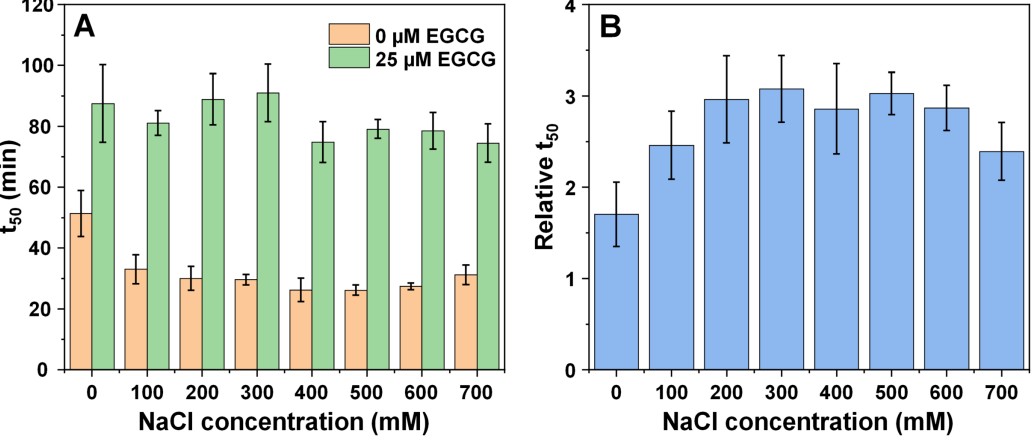

**Figure 3** **Amyloid-beta (1–42) (2 μM) aggregation half-time ($t_{50}$) (A) and relative $t_{50}$ values (B) in the absence or presence of 25 μM EGCG.** Relative $t_{50}$ values were obtained by calculating the ratio between average $t_{50}$ of samples with EGCG and average $t_{50}$ without EGCG. For each condition, half-time values were calculated from six repeats, error bars are for one standard deviation. All raw kinetic data is available as Supplemental Information.               

dependence on the square root of NaCl concentration was distinct from both the insulin and alpha-synuclein cases. There was a positive trend up to 300 mM NaCl, followed by a linear decrease at increasing ionic strength conditions. This indicates that the charge screening effect of NaCl (up to 300 mM) had a positive effect on the lag time values (Fig. S6). In the case of amyloid beta, it was not possible to obtain high quality FTIR spectra due to the low peptide concentration (2 μM). Using a higher amount of each sample in the described FTIR method or exchanging the fibrils into $D_2O$ with 10 kDa concentrators also did not yield sufficient quality results. Such a low concentration and the presence of a relatively high amount of EGCG also prohibited the use of other fibril characterization methods, such as CD spectroscopy or dye-binding assays.

Since it is known that the solution's ionic strength may influence fibril and amyloidophilic molecule interactions, the observed results could be caused by several ionic-strength-related factors, such as EGCG-fibril affinity and interaction or competitive binding with ThT. In order to examine if different NaCl concentrations have an effect ThT and EGCG interactions, the dye molecule absorbance spectra were scanned when they are separately (after which the spectra were combined) and together, as described in the Materials and Method section. When the combined spectra of ThT and EGCG are compared to the spectra of ThT-EGCG mixtures under 100 mM and 800 mM NaCl conditions, there do not appear to be any major differences (Figs. S7A and S7B). A change in their interaction would likely be seen as a shift in spectra maximum positions or lower/higher absorbance at their characteristic bands, however, this is not the case.

To test the competitive binding hypothesis, both ThT and EGCG were mixed with insulin fibrils under 100 mM and 800 mM NaCl conditions (separately and together), after which the fibril samples were centrifuged and their supernatant absorbance spectra were scanned. In the case of 50 μM ThT and 50 μM EGCG, we observed a lower absorbance value at 412 nm (characteristic maximum ThT absorbance peak) and an increase in

absorbance at 300 nm (position where both EGCG and ThT absorb light) when comparing low and high ionic strength conditions (Figs. S7C and S7D). A similar effect was seen when 100 µM ThT and 100 µM EGCG were used (Figs. S7E and S7F). This indicates that higher ionic strength leads to more bound ThT molecules (which is known to be the case for insulin and alpha-synuclein (*Mikalauskaite et al., 2020*)) and also results in less bound EGCG. These differences, however, are not significant enough to constitute this process as the sole reason for the loss of EGCG inhibitory potential at higher NaCl concentrations.

## DISCUSSION

Taking all of these results into consideration, it is quite clear that ionic strength plays a major role in both amyloid aggregation, as well as determining the inhibitory potential of EGCG. However, the interplay between ionic strength, EGCG and amyloid proteins appears to be a lot more complex than one would initially assume. The first interesting aspect is that a different concentration of EGCG was required for each protein/peptide to achieve a 3 to 4-fold increase in relative $t_{50}$ values. For insulin (pI ~ 5.3, condition pH = 2.4) the protein/EGCG ratio was 8:1, while for alpha-synuclein (pI ~ 4.7, condition pH = 7.4) it was 1:1. Despite an eight times different ratio, the highest increase in $t_{50}$ values was 4-fold (with similar $t_{50}$ values), indicating that either protein charge or the type of aggregation intermediates may play an important role in the inhibition process. Conversely, amyloid-beta (pI ~ 5.8, condition pH = 7.0) required a 0.08:1 ratio to even come close to similar inhibition. This may be due to the relatively fast aggregation of amyloid-beta when compared to both other proteins, thus significantly reducing the possibility of EGCG inhibiting the process.

The second factor is the effect that NaCl concentration has on fibril secondary structure. It is a known fact that the solution's ionic strength can lead to the formation of distinct fibril conformations (*Bousset et al., 2013*; *Gaspar et al., 2020*) and the potential inhibitor molecule may have a weaker or stronger effect, based on its interaction with the aggregate. In this case, we observe both gradual (alpha-synuclein) and quick (insulin) fibril structural variations with increasing ionic strength. The second factor is that EGCG, by itself, induces such structural variations, as seen in the case of insulin fibrils at 100 mM NaCl. Since both NaCl and EGCG can influence aggregate conformations, these factors are likely at play simultaneously, which increases the overall complexity of the fibrillization process.

Another interesting aspect is that there seems to be an optimal ionic strength, which showcases the best inhibitory potential of EGCG. Based on relative half-time values, low ionic strength conditions favour the strongest inhibition for insulin and alpha-synuclein, while for amyloid-beta (1–42) it appears to be almost identical at all NaCl concentrations above 0 mM. Conversely, certain conditions reduce the potency of EGCG to such a point, that it may almost appear as ineffective. Examining the charge screening of NaCl for each protein, it seems that alpha-synuclein is less susceptible to the screening effect when compared to insulin, while for amyloid-beta, an initial increase in ionic strength slightly improved the peptide-EGCG interaction (Fig. S6). Taking into account that this is just one

out of a whole plethora of potential anti-aggregation compounds, it is very likely that other compound effectiveness also depends on the solution's ionic strength. This leads to the possibility of certain drug molecules being classified as not having any effect on amyloid aggregation if the reaction conditions are not suitable for them specifically. A similar observation was also made with EGCG and environmental conditions, where the compound was only potent at certain solution pH values (*Sternke-Hoffmann et al., 2020*).

We also have to discuss the possible role that ionic strength has in determining EGCG effectiveness. The first possibility is that the increasing NaCl concentration makes it difficult for EGCG to interact with either the native state protein or the various aggregation intermediates through electrostatic effects. This does not appear to be an issue when it interacts with fibrils, as both 100 mM and 800 mM NaCl led to a similar number of bound EGCG molecules. The second possibility is that EGCG is only effective against certain types of aggregation intermediate species, as such interaction with intermediates has been shown previously (*Andrich & Bieschke, 2015*). If ionic strength changes what type of nuclei or protofibril species form, inhibition may not occur for the different type of aggregate. This does not explain, however, why EGCG seems to also change the conformation of aggregates, unless it does so by preventing certain nuclei from forming. The third probable cause is related to the aggregation kinetics. In general, amyloid fibril formation proceeds slower at low ionic strength conditions, as seen for all three proteins/peptides used in this study. If new nuclei appear less frequently, then the same concentration of EGCG would have a higher chance of preventing their formation and subsequent elongation into fibrils. It is also possible that multiple factors are at play simultaneously, resulting in a highly complex environment-based inhibitor-protein interaction.

## CONCLUSIONS

These results show that each amyloidogenic protein/peptide may have specific ionic strength conditions where they are most susceptible to the inhibitory effect of EGCG. It appears that the inhibitor is generally most potent at lower ionic strength conditions for insulin and alpha-synuclein and an increasing concentration of NaCl reduces its effectiveness. This is not the case for amyloid-beta (1–42), where inhibition is quite similar under all ionic strength conditions above 100 mM NaCl. In addition, the interplay between ionic strength and the inhibitor molecule is highly complex, which affects multiple aggregation parameters, such as aggregation kinetics and fibril conformations. Considering that this is just one of the multiple possible variables in protein aggregation studies, it showcases the importance of taking into account every environmental factor during amyloid formation.

### Funding

This research was funded by the grant no. S-SEN-20-3 from the Research Council of Lithuania. The funders had no role in study design, data collection and analysis, decision to publish, or preparation of the manuscript.

## Grant Disclosures

The following grant information was disclosed by the authors:
Research Council of Lithuania: S-SEN-20-3.

## Competing Interests

The authors declare that they have no competing interests.

## Author Contributions

- Mantas Ziaunys conceived and designed the experiments, performed the experiments, analyzed the data, prepared figures and/or tables, authored or reviewed drafts of the paper, and approved the final draft.
- Kamile Mikalauskaite performed the experiments, authored or reviewed drafts of the paper, and approved the final draft.
- Andrius Sakalauskas performed the experiments, authored or reviewed drafts of the paper, and approved the final draft.
- Vytautas Smirnovas conceived and designed the experiments, analyzed the data, authored or reviewed drafts of the paper, and approved the final draft.

## Data Availability

The raw data is available in the Supplemental Files.

## Supplemental Information

Supplemental information for this article can be found online at http://dx.doi.org/10.7717/peerj.12381#supplemental-information.

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
