# Peer review of "Interplay between epigallocatechin-3-gallate and ionic strength during amyloid aggregation"

_PeerJ, doi:10.7717/peerj.12381_

## Round 0.1 · original submission · Major Revisions

All three reviewers have made a detailed list of items that need to be addressed in your response and revised manuscript.

Staff Note: Reviewer 1 has suggested that you cite specific references. You may add them if you believe they are especially relevant. However, we do not expect or require you to include these citations, and if you do not include them, this will not influence our decision.

Reviewer 1 ·

Basic reporting

The paper is well written and with a clear language making the main results easy to understand.
In the introduction the literature seems to be a little poor. There are several studies involving the interactions of natural molecules with amyloid fibrils, I suggest to include some more generic regarding this topic (lines 29-30) such as:
https://doi.org/10.1371/journal.pbio.1001080
https://doi.org/10.1039/C0NP00027B

Lines 42-45: the effect of EGCG on amyloid have been demonstrated recently also by computer studies such as:
https://doi.org/10.1016/j.jmgm.2017.06.018
https://doi.org/10.3390/ijms21155462

When talking about the ionic strength, authors should include what is the range of ionic strength in the human body to have a comparison with values in the paper.

Figures in the main text are clear and easy to understand. Figure S2, S3 and S4 are clear, but the x axis starts from a negative value of time and it should be better to start from 0. Moreover, the maximum timescale reached seems to be in only one case 1800 min, while all other graphs stop earlier. Is it possible to have an x scale going from 0 to 2500 min?

Experimental design

The methods employed to obtain the results are well described. About the amyloid-beta (lines 134-136), why the quality is not enough? What is the main problem and why authors did not solved it?
However, methods are correctly described and results are reproducible.

The study of the effect of the ionic strength on the interaction of EGCG with amyloid fibrils is of great interest to understand the effect of drugs on the protein aggregation. Moreover, the competition between ThT, EGCG and ionic strength is of fundamental importance to design new and more effective drugs. In this paper, authors employed a high variety of values for the ionic strength.

Validity of the findings

The novelty of this paper rely on the study of the effect of the ionic strength on the binding of EGCG with amyloid fibrils. Conclusions are supported by results and the high number of replicas, data and controls make the results robust.

Additional comments

The conformational changes of insulin is very interesting, but I did not found something similar in literature. Please, check if there are other papers showing that the increase in the ionic strength could alter and in which manner the secondary structure of insulin. Naturally, results regarding the interaction of EGCG with insulin at high ionic strengths should be carefully explained.

The competition of ThT and EGCG is interesting, but a brief discussion at the molecular level based on previous studies should be included.

In the conclusion (line 261), authors should give the value of the optimal ionic strength for the three proteins studied.

Reviewer 2 ·

Basic reporting

Minor comments

Line 21: change ‘several’ to ‘many’
Line 26: It is poor grammar to start a sentence with ‘Which …’
Line 37: ‘course of aggregation’ should include the rate of aggregation
Line 118: change ‘protein’ to ‘peptide’

Lines 59-62: The use of inherently fluorescent compounds as potential inhibitors of amyloid fibril formation can also give rise to false positives in assays when using dyes such as thioflavin T (Hudson et al. (2009) FEBS J. 276, 5960).
The figure captions need to include the concentration of the peptide/ protein used, not just the concentration of EGCG.

Experimental design

This manuscript describes an experimental investigation of the effect of ionic strength (via NaCl concentration variation) on amyloid fibril formation of three common fibril-forming species peptide (amyloid beta) and two proteins (insulin and a-synuclein) in the presence of the well-known polyphenol fibril inhibitor, EGCG. Teasing out an accurate understanding of what is occurring at the molecular level for each of the fibril-forming species, with many variables present, is not easy. Below are suggestions on how to improve the manuscript:

Line 92: At what pH were the aggregation experiments conducted with insulin? Presumably, it was acidic, i.e. a pH around 2 as insulin readily forms amyloid fibrils under these conditions.
Line 111: Was amyloid beta 1-40 or amyloid beta 1-42 used for the experiments? The latter aggregates much faster because of the additional two hydrophobic residues. From the relatively short t50 values in the data presented in Figure 3, it would seem to be amyloid beta 1-42.
Line 122: Why were the FTIR spectra acquired in D2O?
Why were the data for t50 selected for analysis? How about also analyzing the data for ThT fluorescence intensity at the end of the experiment, the time for onset of aggregation (the lag phase) and the initial rate of aggregation (i.e. kinetics of aggregation), which are all standard parameters to investigate inhibition efficiency? The supplementary data show a large amount of variation in the onset of aggregation (i.e. the lag phase) for the three target species in the presence of low concentrations of NaCl (up to ~ 300 mM) but greater consistency above this concentration. What is the explanation for this? Significant variation also occurs in Supplementary Figures 2-4 in the fluorescence intensity at the end of the experiment for all NaCl concentrations.
For all three fibril-forming species interacting with EGCG, why were the particular ratios chosen, e.g. 25 uM EGCG : 200 uM insulin, i.e. a 1:8 molar ratio? For a-synuclein and amyloid beta, the ratios were very different (1:1 and 1:0.08). How much does EGCG inhibit insulin, a-synuclein and amyloid beta fibril formation at these ratios? The data in the Supplementary Figures 2-4 imply that intermediate levels of inhibition of aggregation by EGCG are present for all three species. This needs to be accurately quantified and compared for all three species in order to interpret the data accurately.
From the data presented in all three figures (and the supplementary figures) in the absence of EGCG, amyloid beta aggregates much faster than insulin and a-synuclein, which is most likely a contributing factor to the inefficiency of the EGCG to affect amyloid beta aggregation, unless it is present in large excess. In general, small (e.g. EGCG) and large molecule inhibitors are much more efficient at preventing fibril formation of slowly aggregating species.
In Figure 3 (amyloid beta + EGCG), why are the data for 0 mM NaCl presented, but they are not reported for insulin and a-synuclein interacting with EGCG? There is little variation in t50 in Figure 3 for all concentrations of NaCl.
Lines 195-197: If a-synuclein did not aggregate in the presence of EGCG and no salt, then this implies that EGCG totally inhibited a-synuclein aggregation under these conditions. Importantly for such a conclusion, did a-synuclein aggregate on its own, i.e. in the absence of EGCG and salt?
Lines 181-186: In general, EGCG, like other fibril inhibitors, stabilizes an intermediately folded form of insulin (one early on its unfolding, fibril-forming pathway), which is composed of some beta-sheet. It does not have as much beta-sheet as the amyloid fibrillar form which adopts a cross beta-sheet conformation.

Validity of the findings

Consideration should be given in the Discussion of the effect of charge on the three fibril-forming species, i.e. their pI values. Clearly these, along with pH of the experiments, will have a major effect on interaction with Na and Cl ions, particularly for a-synuclein and amyloid beta as they are unstructured entities and therefore their calculated pI values are unaffected by secondary and tertiary structural considerations. Insulin is mainly alpha-helical in its native state but partially unfolds and dissociates from its hexameric form at low pH to facilitate its fibril formation.
Lines 281-283 in the Discussion: In general, higher concentrations of NaCl (i.e. higher ionic strength) lead to greater screening of electrostatic interactions between the peptide/ proteins concerned and a consequent shortening of the t50, i.e. promotion of aggregation. At low ionic strength and less consequent screening, repulsion between the peptide/ proteins leads to reduced aggregation, and longer t50 times.

Reviewer 3 ·

Basic reporting

The authors do a good job in general of reporting their work which is communicated in good English and with a sensible and logical flow.
One practical point: please mention both protein and EGCG concentrations (both in mM) in the legends to all figures to allow comparison.

Experimental design

1. The work is critically dependent on the use of plate reader data. I commend the authors for considering positional effects in their plate setup (Fig. S1). Strictly speaking one could randomize it even further but that becomes complicated to administrate. For good measure the authors should describe the design of the plate content (the first and second halves of the plate are identical).
2. Why do the authors use 100 µM EGCG with aSN and only 25 µM with insulin and Abeta? This makes it difficult to compare the three proteins. It also raises the thorny issue of concentration dependence in general, i.e. can the effect of salt be overridden by higher [EGCG]?

Validity of the findings

I have some issues with the authors’ interpretations of their results as elaborated in the next section.

Additional comments

1. My main issues relate to the statistical significance of the conclusions drawn from the data.
a. Fig. 2B: I am somewhat skeptical of the conclusions drawn from this figure in view of the relatively large errors (which are inevitable in view of the stochasticity of EGCG effects, cfr. above). At the very least, the authors should subject their values to a more thorough statistical analysis. Currently I would conclude that there is no significant change between values at 100-400 mM NaCl.
b. Similarly, the values in Fig. 3B in my view show similar values at 200-700 mM NaCl; the value at 100 mM is lower but the associated errors make the significance questionable. I disagree with the conclusion that “The relative t50 values (Figure 3B) followed a similar trend as in the case of alpha-synuclein.” – unless that trend is revised to say that that there is no trend (which is perhaps a little harsh).
c. The authors state that “The decreasing t50 values also greatly reduced the stochasticity of spontaneous aggregation, which was quite large at lower ionic strength conditions (Supplementary Figure S2).” This is only the case for EGCG; in the absence of EGCG, the data are (enviably) reproducible at all [NaCl] and this should be clarified. Similar observations are seen in Fig. S3 for alpha-synuclein. It is intriguing that EGCG makes the aggregation process more random at low [NaCl] and the authors are encouraged to speculate on this in the Discussion.
2. In general I am also puzzled by the weak effect of EGCG on aggregation of these proteins. In my experience (admittedly using EGCG that had not been pre-oxidized), 1:1 molar ratios of EGCG are sufficient to suppress aggregation completely. The authors are requested to comment on this. Also, the oxidation change will affect the types of interactions between EGCG and protein. It would have been good to compare with unoxidized EGCG which most researchers use anyway.
3. Overall I think it is an important message that EGCG has the greatest effect at low [NaCl] and this is the safest conclusion that can be drawn. The high quality of the ThT time profiles obtained by the authors makes it possible to make more detailed quantitative conclusions. E.g. they should plot t(lag) versus the square root of [NaCl] to check for screening effects and make the appropriate conclusions.
Minor issues:
4. The EGCG oxidation step seems sensible enough, but do the authors have a way to validate the completeness of the oxidation (e.g. by spectroscopy)?
5. Fig. 1B and 2B: Define the relative t(lag) (ratio between t(lag) with and without EGCG).
6. Fig. 1CF and 2CF: describe the positions of the two stippled lines representing beta-sheet structures and carbonyl groups (and label them on the graphs).

---

## Round 0.2 · accepted · Accept

Together with the reviewers, I am pleased with the revised version of your manuscript.

Reviewer 1 ·

Basic reporting

Authors replied to all my concerns and I think that the paper has improved inn quality

Experimental design

All experiments are well described and they can be easily reproducible.

Validity of the findings

In the new version of the manuscript, all data are available and results are correctly described. Conclusions are supported by results and they clearly indicate the most important findings of this work.

Reviewer 3 ·

Basic reporting

Pass

Experimental design

Pass

Validity of the findings

Pass

Additional comments

The authors have satisfactorily addressed the issues I originally raised.